# Isolation and characterization of halophilic and halotolerant fungi from man-made solar salterns in Pattani Province, Thailand

**Lakkhana Kanhayuwa Wingfield** *, Ninadia Jitprasitporn, Nureeda Che-alee

Division of Biological Science, Faculty of Science, Prince of Songkla University, Hat Yai, Songkhla, Thailand

* Lakkhana.k@psu.ac.th

**Data Availability Statement:** All relevant data are within the paper and its Supporting Information files.

## Abstract

The present study explored culturable halophilic and halotolerant fungi from man-made solar salterns in Pattani Province, Thailand. A total of 24 fungal isolates were discovered and characterized using morphological and molecular identification. Production of extracellular enzymes, secondary metabolites and mycoviruses was examined. Growth was observed in salinity and temperature ranges between 0%-20% and 28–40˚C, respectively. Growth in different environmental conditions confirmed the halophilic or halotolerant nature of some strains. Fungal isolates were phylogenetically classified into seven different genera belonging to *Aspergillus*, *Cladosporium*, *Curvularia*, *Diaporthe*, *Ectophoma*, *Fusarium* and *Penicillium*. An enzymatic production test revealed that thirteen isolates could produce proteases and amylases at different levels. The presence of mycoviruses was detected in three isolates. Seventeen of the 24 isolates produced antimicrobial metabolites. The majority of these active isolates were identified as *Aspergillus* and *Penicillium* species. Crude extracts of the fungal mycelia and culture broths from these isolates had an inhibitory effect on both Gram-positive and Gram-negative bacteria and human pathogenic fungi. Research into fungi from saline environments could reveal fungal strains of biotechnological and industrial interest.

## Introduction

Fungi are eukaryotic organisms that can be either unicellular yeast or multicellular hypha. They are mainly found in soil but also in aquatic habitats. Fungi have been observed in extreme environments such as the polar regions, hot springs, deserts, saline water, ocean pits and very low pH environments [1]. They are also found in hypersaline environments, which usually contain salt concentrations higher than the 3.5% typical of seawater [2]. Examples of hypersaline environments are solar salterns and crystallization ponds. In these systems, the evaporation of water in this environment causes precipitation of minerals and sodium chloride (NaCl) at levels >300 practical salinity unit (psu), turning neutral pH soil slightly alkaline [3].

Organisms that require salt for their growth are known as halophiles. Halophiles can be categorized into five major groups: 1) non-halophiles, which grow in <0.2 M (*c.a.* 1%) NaCl; 2)

**Funding:** This work was financially supported by Office of the Permanent Secretary, Ministry of Higher Education, Science, Research and Innovation (Grant No. RGNS 63-239); and the National Science, Research and Innovation Fund (NSRF) and Prince of Songkla University (Grant No. MED6505096g). The funders had no role in study design, data collection and analysis, decision to publish, or preparation of the manuscript.

**Competing interests:** The authors have declared that no competing interests exist.

mild halophiles, which require 0.2–0.5 M (*c.a.* 1–3%) NaCl; 3) moderate halophiles, which require 0.5–2.5 M (*c.a.* 3–15%) NaCl; 4) borderline extreme halophiles, which require 1.5–4.0 M (*c.a.* 9–23%) NaCl; and 5) extreme halophiles, which require 2.5–5.2 M (*c.a.* 15–32%) NaCl [4]. Fungi that inhabit extreme environments are mainly categorized as belonging to the imperfect stage of the Ascomycota [5]. *Aspergillus* and *Penicillium* are the predominant genera that have been reported to exist in hypersaline environments. Other genera include *Acremonium*, *Alternaria*, *Chaetomium*, *Cladosporium*, *Fusarium* and *Wallemia* [6–12]. New fungal species are frequently found in hypersaline environments, as described in several reports, which include three species of *Wallemia* [13], three species of *Gymnoascus* [14] and twelve species of *Cladosporium* [15].

Salinity can directly affect fungal growth and sporulation. For example, at salinities above 5%, increased sporulation and chlamydospore formation are observed while fewer hyphae are produced [6]. Generally, fungi in extreme environments produce extremolytes and extremozymes to cope with osmotic stress [16]. When environmental osmolarity is high, some fungi can neutralize water loss by accumulating $K^+$ ions in their cells [17]. Other fungi accumulate osmolytes (polyols, sugars and amino acids) as compatible organic solutes [18]. The high salinity of hypersaline environments makes these habitats a potential source of halophilic fungi that can produce enzymes for industrial applications. Investigations of enzymes obtained from halophilic fungi have identified amylases, cellulases, lipases and proteases [19]. In addition, secondary metabolites produced by fungal halophiles have been reported to include antimicrobial agents, toxins and pigments [20]. Many fungal species can be infected with fungal viruses known as mycoviruses [21]. Mycovirus infections can exhibit a range of phenotypes on fungal hosts, both beneficial and harmful. It has been demonstrated that mycoviruses can confer both thermal tolerance and thermal stress on a fungus. For instance, infection by the double-stranded (ds)RNA mycovirus Curvularia thermal tolerance virus (CThTV) induces thermal tolerance in the endophytic fungus *C. protuberata* and its host plant *Dichanthelium lanuginosum* [22]. In addition, a mycovirus can induce the production of the metabolite tenuazonic acid in a rice blast fungus *Magnaporthe oryzae* [23].

Studies of the physiology and genetics of halophilic fungi are significant to the understanding of the ecological roles and potential biotechnological applications of their enzymes and metabolites, and mycoviruses. Several halophilic fungi have been isolated from solar saltern ecosystems and described. However, their diversity, ecological role and potential to produce secondary metabolites are still poorly understood. In Thailand, saline soil areas are primarily found along the coastal areas and in a northeast inland region, that covers an area of 130 km$^2$. Man-made solar salterns constructed for the extraction of NaCl are mainly located along the Central Region coastal belt in the provinces of Phetchaburi and Samut-sakorn. Some halophilic fungi from these habitats have been described [24], but there is no record of halophilic fungi from habitats in the coastal belt of the Southern Region of Thailand. In the present study, we report the first study to consider the diversity of fungal halophiles in solar salterns in Pattani Province, southern Thailand. We discovered fungal phylotypes that could produce extracellular enzymes and bioactive metabolites, and mycoviruses to be further used for biotechnological and industrial interests. The salt tolerance and temperature tolerance of the isolates were also evaluated.

## Materials and methods

### Sampling sites

Soil samples were collected at the end of dry season (May 2020) from solar salterns located in the province of Pattani in southern Thailand (6˚88'21.5"N 101˚28'04.5"E). Sampling was

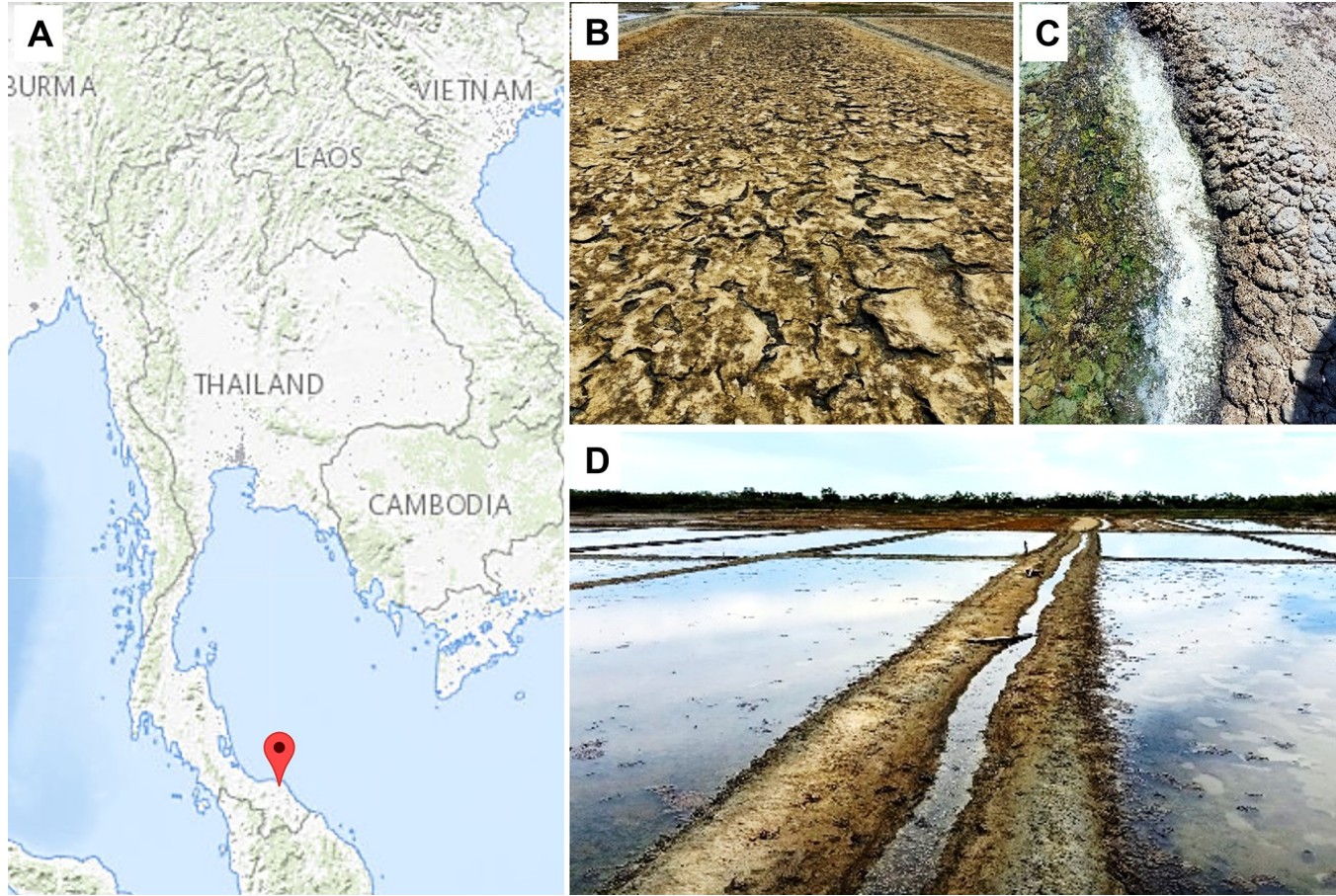

**Fig 1. Sampling sites.** A) Geographic location of the sampling sites (USGS National Map Viewer (http://viewer.nationalmap.gov/viewer/)). B-D) Characteristics of the sampling sites and neighboring locations at Ban Bana district, Pattani Province, Thailand.

carried out at two sites, which are presented in Fig 1. The man-made salterns lie about 1 m above sea level, and the air temperature at the sampling site was recorded as 34.0˚C. Soil samples were collected from the surface layer of the soil at depth of 1 to 5 cm and from the subsoil at a depth of 5 to 15 cm. The collected samples were placed in sterile collection bags and transported to the laboratory.

## Physicochemical analysis of soil

Soil suspensions were prepared for physicochemical analysis by mixing 10 g of soil sample in 50 mL of distilled water. The salinity and pH of soil suspensions were determined using a salinity refractometer and pH meter, respectively. Moisture content was measured by calculating the weight difference between fresh and dried soil samples. Organic matter and total organic carbon (TC) were respectively determined by the Walkley-Black wet combustion method and a total organic carbon analyzer (Teckmar-Dohrmann Phoenix 8000) [25]. Total nitrogen (TN) was measured by the Kjeldahl digestion method using a Tecator DS-6 Digester [26].

## Isolation and morphological identification of fungi

Fungal isolation was performed using a ten-fold serial dilution method on potato dextrose agar (PDA) supplemented with 10% NaCl and 100 μg/mL chloramphenicol. A mixture of 10 g

of soil in 90 mL of sterile phosphate buffer was shaken at 200 rpm for 30 min and serially diluted at $10^{-1}$ to $10^{-3}$. One hundred μL from each dilution were spread plated, and the plates were incubated at 28˚C for 7–21 days. Axenic fungal cultures were obtained by transferring isolated hyphal tips to fresh PDA plates containing the same salt concentrations, for incubation at 28˚C for one to two weeks. Fungal isolates were identified as different genera and species by observing their macroscopic morphology (growth rate, colony color, pigmentation and colony surface) and microscopic morphology (sexual and asexual reproductive organs, hyphae and ultrastructure). The identification keys of Samson et al. [27], Dugan [28] and Ellis et al. [29] were used to identify isolates in different genera.

## Molecular identification

Fungal DNA was extracted as previously described by Wingfield and Atcharawiriyakul [30]. Briefly, fungal mycelium frozen in liquid nitrogen was ground, and 0.1 g of the ground mycelium was subjected to an IRRI DNA extraction method. A volume of 1 mL of extraction buffer [50 mM Tris-HCl (pH 8), 25 mM EDTA, 300 mM NaCl, 1% SDS] was added, followed by the addition of 500 μL of chloroform/isoamyl alcohol (24:1). The mixture was then centrifuged at 13,000 rpm for 1 min. A 750 μl supernatant was transferred and subjected to ethanol precipitation. The resulting DNA pellet was washed with 70% ethanol and resuspended in 100 μl of TE buffer (10 mM Tris-HCl, 1 mM EDTA pH 8.0). DNA integrity was checked by electrophoresis in 1% agarose gels, followed by staining with ethidium bromide (10 mg/mL). The purity of DNA was assessed from the ratio of absorbance at 260 and 280 nm.

PCR amplification of internal transcribed spacer (ITS) regions of fungal rDNA was performed using ITS primer sets (ITS1/ITS4N and ITS4N/ITS5) as described by Wingfield and Atcharawiriyakul [30]. In addition to the ITS regions, other DNA regions were amplified according to the fungal genera. β-tubulin was amplified for strains of Aspergillus and Penicillium using primers Bt2a/Bt2b [31], the 1α translation elongation factor gene (TEF-1α) was amplified for strains of Fusarium and Curvularia using primers EF1F/EF1R [32] and actin was amplified for strains of Cladosporium using primers ACT-512F/ACT-783R [33]. All the above PCR amplifications were done in a 50 μL reaction mixture containing 0.2 μL Taq DNA polymerase (5 U/ μL, Invitrogen, USA), 5.0 μL 10X PCR buffer, 1.0 μL dNTP mixture (10 mM), 1.5 μL $MgCl_2$ (50 mM) and 2.5 μL primer mix (10 μM each). The PCR conditions were as follows: an initial denaturation at 94 ˚C for 5 min, followed by 35 amplification cycles of denaturation at 94 ˚C for 45 sec, annealing at 55 ˚C for 30 sec and extension at 72 ˚C for 90 sec, for completion with a final extension at 72 ˚C for 10 min. All reactions were performed using a thermocycler (USA). Aliquots of 5 μL of the PCR products were resolved in 1% agarose gels in 1X tris acetate ethylenediaminetetraacetic acid (TAE) buffer and visualized under UV light. The PCR products were purified using the MinElute® Gel Extraction Kit (QIAGEN) as described by the manufacturer. The purified amplicons were sent for sequencing at Macrogen Inc., South Korea, and the sequences obtained from each fungal isolate were further analyzed using the Basic Local Alignment Search Tool (BLAST) at the National Center of Biotechnology Information (NCBI) server.

## Physiological and biochemical characterization

Physiological characterization was performed to determine the ability of the fungal isolates to grow in the presence of different NaCl concentrations ((w/v) 0%, 5%, 10% and 20%) and in different temperature ranges (25˚C and 40˚C). Growth was scored depending on the diameter of the colony after 7 days of incubation or after 15 days of incubation for slow-growing isolates.

The ability of the isolates to produce extracellular enzymes was examined. Amylase activity was tested on nutrient agar containing 2 g/L of soluble starch. Cultures were incubated for 2 to 5 days, depending on the growth rate of each isolate, and then flooded with an iodine solution. A clear zone around the colony indicated the presence of amylase [34]. Cellulase activity was tested on a solid medium containing 1% cellulose (7.0 g/L $KH_2PO_4$, 2.0 g/L $K_2HPO_4$, 0.1 g/L $MgSO_4.7H_2O$, 1.0 g/L $(NH_4)_2SO_4$, 0.6 g/L yeast extract, 10 g/L microcrystalline cellulose and 15 g/L agar) [35]. After incubation, the cultures were further incubated at 50˚C for 16 h to accelerate enzyme activity. The cultures were then flooded with 5 mL of 1% Congo red solution and rinsed with distilled water to detect a hydrolysis zone. Protease activity was tested on casein agar medium containing 30% skim milk and 2% agar. Degradation of casein was indicated by a clear zone around the colony [36]. Lipase activity was tested on a solid medium containing tween 80 (10 g/L peptone, 5 g/L NaCl, 0.1 g/L $CaCl_2.2H_2O$, 17 g/L agar and 10 mL/L Tween 80). Tween 80 was sterilized before addition to the sterile medium. Cultures were placed at 4˚C for 12 h after incubation to observe opaque precipitation around the colony [34]. Each enzyme activity was evaluated by an enzymatic index (EI) where EI = R/r (R being the diameter of the clear zone and r the diameter of the colony) [32]. Mycovirus dsRNAs were extracted using LiCl fractionation as described by Hull and Covey [37]. dsRNA samples were purified by phenol-chloroform extraction. DNA and ssRNA contamination were removed by sequential DNase I and SI nuclease treatment. The viral genome was proved to be dsRNA by digestion with RNase III. The presence of the dsRNA was checked by electrophoresis in a 1% (w/v) agarose gel in 1X TAE buffer.

## Antimicrobial assay

Fungal isolates were grown in a potato dextrose broth (PDB) for 21 days at 28˚C under static conditions. Culture broths of 80 μL were used for primary screening of antimicrobial activity by agar well diffusion method [38] against pathogenic bacteria and fungi. Five Gram-positive bacteria, *Enterococcus faecalis*, *Micrococcus. luteus* (ATCC9341), *Staphylococcus. aureus* (ATCC25923), *S. epidermidis*, and methicillin-resistant *S. aureus* (MRSA); three Gram-negative bacteria, *Escherichia coli* (ATCC25922), *Pseudomonas aeruginosa* (ATCC27853), and *Salmonella* Typhi (ATCC19430); a pathogenic yeast, *Candida albicans* (ATCC90028); and two human pathogenic fungi, *Aspergillus fumigatus* AF293 and *Microsporum canis*, were tested. The culture broth of each fungal isolate was placed on a plate previously spread with a bacterial suspension (Mueller-Hinton Agar, MHA) or a yeast or fungal spore suspension (Sabouraud dextrose agar, SDA). A hyphal inhibition test [39] was also performed against the *A. fumigatus* AF293 and *M. canis*. The broad-spectrum antibiotic chloramphenicol (50 μg/mL) and the antifungal amphotericin B (50 μg/mL) were used as positive controls for bacteria and yeast, respectively. After incubation at 35 ˚C for 18 h for bacteria, 28 ˚C for 24–48 h for yeast and 28 ˚C for 48 h for filamentous fungi, inhibition zones were measured as the mean diameter of the 8 mm wells plus the clearing zone in triplicates. The values obtained were presented as means ± standard deviation (n = 3).

Fungal isolates showing active antimicrobial activity were selected for secondary screening. Culture broths were filtered to separate filtrates and mycelia, which were both extracted three times with 95% ethyl alcohol. Various concentrations (1–100 mg/mL) of the ethanolic extracts of the filtrates and mycelia were tested for antimicrobial activity using the agar well diffusion method described above. Vancomycin and gentamicin were used as standard antibacterial agents and amphotericin B and miconazole were used as standard antifungal agents. The values obtained were presented as means ± standard deviation (n = 3).

# Results

## Physicochemical analysis of soil

The results of the physicochemical analyses of soil samples are presented in Table 1. Soil samples were mildly alkaline and had a moderate salinity rate, indicating the hypersaline condition of the sampling sites [40]. Clay material was dominant and the soil color ranged from green, and brown to dark gray. The solar saltern soils showed a deficiency in organic matter as total organic carbon, and organic matter levels were low. Nitrogen content was lower than reported in the literature [41].

## Isolation and identification of the fungal isolates

A total of 24 fungal isolates were obtained on PDA supplemented with 10% NaCl (w/v). After plating, the growth of the isolates was monitored and recorded after incubation at 28˚C for periods of between 7 and 21 days. Spore production was observed after 4–7 days for the spore-forming isolates. The isolates varied in macroscopic and microscopic characteristics in terms of colony color, pigmentation, surface and the hyphae were either pigmented or hyaline septate (Fig 2). Twenty-three isolates were identified to the genus level by macroscopic and microscopic observation, while one isolate could not be identified (sterile mycelia). All morphologically identified fungal isolates and the unidentified isolate were selected for molecular identification.

## Molecular identification and phylogenetic analysis

The fungal strains were identified using molecular analysis. BLAST analysis was performed to evaluate the phylogenetic relationships between an individual isolate and its nearest neighbors, and the results were presented as percentage similarity (Table 2). The sequences of most of the 24 assayed strains showed high similarity (>98%) with the sequences deposited in GenBank database.

Among the isolates, ten were affiliated to the genus *Aspergillus* with most of them exhibiting between 96 and 100% similarity. *Aspergillus subalbidus* was the most frequent halophilic fungal species found in the soil samples under study. Other genera were represented in small numbers and showed a close affiliation with known genera in the phylum Ascomycota. They included: *Penicillium* (5 isolates), *Cladosporium* (3 isolates), *Fusarium* (2 isolates), *Ectophoma* (2 isolates), *Curvularia* (1 isolate), and *Diaporthe* (1 isolate). Based on the bootstrap values of the ITS loci (Fig 3), *Aspergillus* and *Penicillium* are sister groups in the same clade and are closely related. The genera *Curvularia*, *Diaporthe*, *Ectophoma* and *Fusarium* fall in the same clade and therefore have a common evolutionary ancestor. Twenty-one strains were affiliated

**Table 1. Physicochemical analyses of soil samples.**

| Parameter | Test result | |
|---|---|---|
| | Site 1 | Site 2 |
| pH | 7.62 | 7.55 |
| Salinity (%w/w) | 3.8 | 3.0 |
| Salinity (g/kg) | 37.53 | 30.42 |
| Moisture content (%) | 27.73 | 34.26 |
| Total nitrogen content (%w/w) | 0.10 | 0.15 |
| Total organic carbon (%w/w) | 1.06 | 1.54 |
| Organic matter (%w/w) | 1.82 | 2.01 |

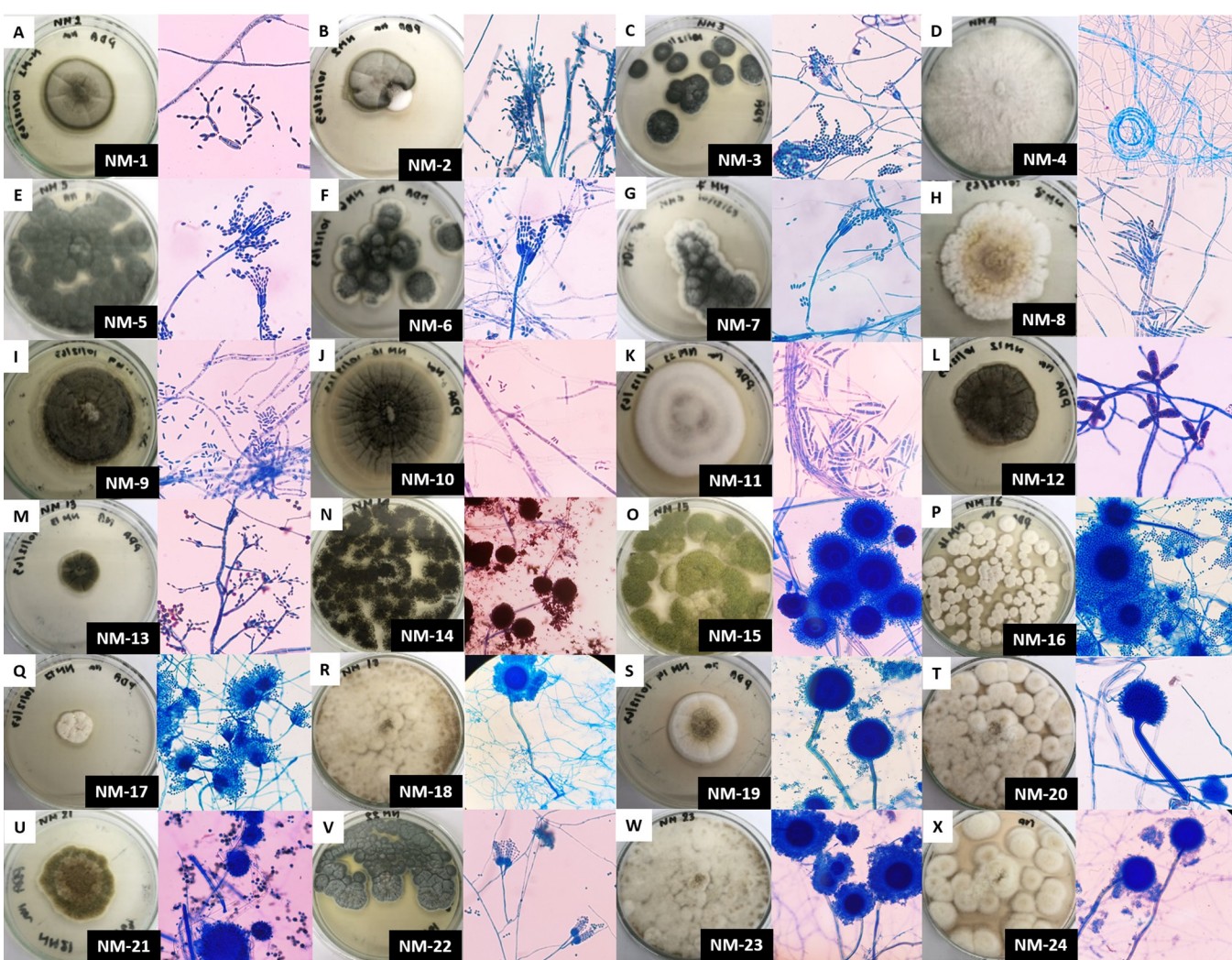

**Fig 2. Morphological characteristics of the isolated fungi.** Morphological observation of the 24 fungal isolates based on macroscopic and microscopic morphology (magnification x40). Colony morphology (left panel), microscopic morphology (right panel). All isolates were grown on PDA plates for 7–14 days at 28°C.

to 13 species belonging to seven genera and five orders. The other three strains were identified only at the genus level, belonging to the genera *Aspergillus* (NM-17, NM-21) and *Cladosporium* (NM-13), showing high similarity (96–99%) with sequences of more than one species. These strains could represent novel species; however, in-depth molecular and phylogenetic studies should be done to confirm their correct taxonomic position.

## Physiological characteristics

Halotolerance and temperature were studied to better understand the growth behavior, adaptation and survival of the fungal halophiles. The results are presented in Table 3 and S1 Fig. The halotolerance test revealed that all fungal strains showed growth on PDA plates supplemented with different NaCl concentrations (0%, 5%, 10% and 20% (w/v)). All isolates could grow on PDA medium without NaCl, indicating the non-obligatory halophilicity of all isolates. With the exceptions of NM-4, NM-6, NM-9, NM-10, NM-11 and NM-12, optimal growth was observed at NaCl concentrations of 5% and 10%. In a salt concentration of 10%, most of the

**Table 2. BLAST analysis results of the fungal isolates from solar salterns in Pattani Province and their close relatives.**

| Isolate code | Locus | Closest relative (BLAST) | Accession number | Identity (%) |
|---|---|---|---|---|
| NM-1 | actin | *Cladosporium tenuissimum* XCSY3 | MT154174.1 | 99.49 |
| NM-2 | actin | *Cladosporium tenuissimum* XCSY3 | MT154174.1 | 96.91 |
| NM-3 | ITS | *Penicillium citrinum* CF18 | KM979730.1 | 99.07 |
| NM-4 | ITS | *Diaporthe miriciae* BRIP 56918a | KJ197284.1 | 99.08 |
| NM-5 | ITS | *Penicillium oxalicum* MLG | OL614490.1 | 99.65 |
| NM-6 | ITS | *Penicillium oxalicum* RCEF4908 | HM053477.1 | 99.82 |
| NM-7 | ITS | *Penicillium oxalicum* RCEF4908 | HM053477.1 | 99.62 |
| NM-8 | ITS | *Fusarium equiseti* NRRL26419 | NR_121457.1 | 99.62 |
| NM-9 | ITS | *Ectophoma pomi* NTUCC17-034 isolate TR_13 | MT112289.1 | 98.60 |
| NM-10 | ITS | *Ectophoma multirostrata* CR32 | MH790211.1 | 99.61 |
| NM-11 | ITS | *Fusarium solani* ALAL-2 | MW923791.1 | 96.26 |
| NM-12 | ITS | *Curvularia pseudobrachyspora* CBS337.64 | MN688815.1 | 99.64 |
| NM-13 | actin | *Cladosporium* sp. A13 | AJ300312.1 | 99.12 |
| NM-14 | ITS | *Aspergillus niger* RMUAN75 | MT550026.1 | 98.99 |
| NM-15 | ITS | *Aspergillus nomius* USPMTOX119 | MK033400.1 | 100.00 |
| NM-16 | β-tubulin | *Aspergillus subalbidus* DTO:129-E3 | KJ775068.1 | 99.61 |
| NM-17 | ITS | *Aspergillus* sp. HF39 | KR081422.1 | 96.15 |
| NM-18 | β-tubulin | *Aspergillus subalbidus* DTO:129-E3 | KJ775068.1 | 99.04 |
| NM-19 | β-tubulin | *Aspergillus subalbidus* DTO:129-E3 | KJ775068.1 | 98.91 |
| NM-20 | β-tubulin | *Aspergillus subalbidus* DTO:129-E3 | KJ775068.1 | 99.12 |
| NM-21 | ITS | *Aspergillus* sp. SFB81 | OL685266.1 | 96.83 |
| NM-22 | ITS | *Penicillium griseofulvum* MSR1 | KJ881374.1 | 98.37 |
| NM-23 | β-tubulin | *Aspergillus subalbidus* CBS 567.65 | MN969366.1 | 99.33 |
| NM-24 | β-tubulin | *Aspergillus subalbidus* CBS 567.65 | MN969366.1 | 98.88 |

isolates showed slow growth while isolates NM-3, NM-8, NM-16, NM-18, NM-19, NM-20, NM-22, NM-23 and NM-24 grew at the same rate as they did in 5% NaCl. However, isolates NM-3, NM-16, NM-18, NM-19, NM-20, NM-23 and NM-24 showed optimal growth at 10%.

The highest salt concentration tested was 20%, which slowed the growth of all tested fungal strains. Only nine isolates (NM-2, NM-16, NM-17, NM-18, NM-19, NM-20, NM-21, NM-23 and NM-24) showed moderate growth, while the rest showed slight growth in 20% NaCl (w/v) (Table 3). Notably, all fungal isolates obtained from the solar salterns could grow across all tested salt concentrations. Halophilic behavior was exhibited by the isolates *Penicillium citrinum* NM-3, *Fusarium equiseti* NM-8, *Aspergillus niger* NM-14, *A. subalbidus* NM-16, *Aspergillus* sp. NM-17, *A. subalbidus* NM-18, *A. subalbidus* NM-19, *A. subalbidus* NM-20, *P. griseofulvum* NM-22, *Aspergillus* sp. NM-23 and *Aspergillus* sp. NM-24 as they were found to grow well in 5 and 10% NaCl (w/v). The remaining isolates were halotolerant. In addition, the study demonstrated that the *Aspergillus* genus is highly tolerant of high salt concentrations. The temperature tolerance test demonstrated that the optimum growth temperature of most isolates was 28˚C. When the temperature was raised to 40˚C, slow growth was observed in all isolates except *D. miriciae* NM-4, which did not grow at all.

## Screening for enzymatic activity

All the isolates were screened for their ability to produce extracellular proteases, cellulases, amylases and lipases by growing on a medium containing the respective substrates. Production of amylases, indicated by positive enzymatic activity on soluble starch, was observed in nine

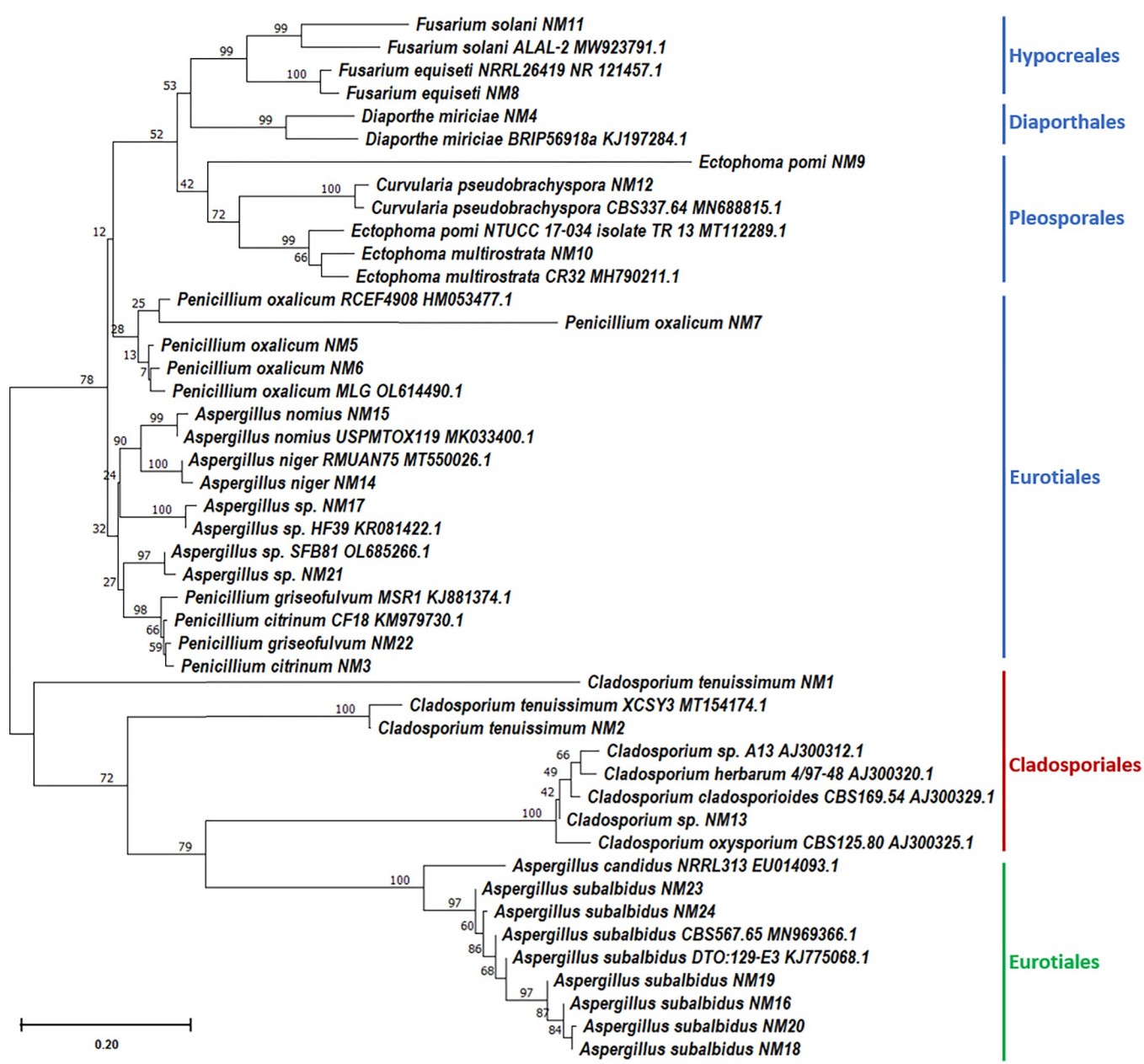

**Fig 3. Phylogenetic analysis of the isolated fungi.** An unrooted phylogenetic tree was generated using the neighbor-joining method based on comparisons of the ITS ribosomal DNA sequences (blue clade), actin (red clade,) and the β-tubulin gene (green clade) of fungal isolates and their closest phylogenetic relatives. Percentages of bootstrap sampling derived from 1000 replications are indicated by the numbers on the tree.

isolates (NM-2, NM-3, NM-5, NM-6, NM-7, NM-16, NM-17, NM-19 and NM-22) (Table 4). Positive protease activity, indicated by a clearance zone around the colony on casein agar, was observed in seven isolates (NM-1, NM-2, NM-3, NM-18, NM-20, NM-22 and NM-24). Eleven out of 24 isolates (46%) did not show any enzymatic activity on any of the substrates, while the production of cellulases and lipases was not detected in any of the isolates.

It was observed that most fungi isolated in this study showed low enzymatic activity, as indicated by clear zone diameters around the colony (EA). Those with a clear zone measuring up to 3 mm were scored as (+). Moderate enzymatic activity for protease production was

**Table 3. Growth characteristics of different halophilic fungal isolates under different physiological conditions (NaCl and temperature).**

| Strain identity | Isolate code | Salt tolerance | | | | | Temperature tolerance | |
|---|---|---|---|---|---|---|---|---|
| | | 0% | 5% | 10% | 20% | Halophile | 28°C | 40°C |
| *Cladosporium tenuissimum* | NM-1 | ++ | +++[1] | ++ | + | Moderate | ++ | + |
| *Cladosporium tenuissimum* | NM-2 | ++ | +++[1] | ++ | ++ | Moderate | ++ | + |
| *Penicillium citrinum* | NM-3 | ++ | ++ | +++[1] | + | High[¥] | ++ | ++ |
| *Diaporthe miriciae* | NM-4 | +++[*][1] | +++ | + | + | Moderate | +++[*] | - |
| *Penicillium oxalicum* | NM-5 | ++ | ++[1] | ++ | + | Moderate | ++ | + |
| *Penicillium oxalicum* | NM-6 | +++[1] | ++ | ++ | + | Mild | ++ | + |
| *Penicillium oxalicum* | NM-7 | ++ | ++[1] | ++ | + | Moderate | ++ | + |
| *Fusarium equiseti* | NM-8 | +++[*] | +++[*][1] | +++[*] | + | High[¥] | +++[*] | ++ |
| *Ectophoma pomi* | NM-9 | +++[1] | +++ | ++ | + | Moderate | ++ | + |
| *Ectophoma multirostrata* | NM-10 | +++[*][1] | +++ | ++ | + | Moderate | ++ | + |
| *Fusarium solani* | NM-11 | +++[*][1] | +++ | ++ | + | Moderate | +++[*] | ++ |
| *Curvularia pseudobrachyspora* | NM-12 | ++[1] | ++ | + | + | Mild | +++ | ++ |
| *Cladosporium* sp. | NM-13 | ++ | ++[1] | ++ | + | Moderate | +++ | + |
| *Aspergillus niger* | NM-14 | ++ | +++[*][1] | ++ | + | Moderate[¥] | +++ | ++ |
| *Aspergillus nomius* | NM-15 | +++[*] | +++[*][1] | ++ | + | Moderate | +++ | ++ |
| *Aspergillus subalbidus* | NM-16 | + | ++ | ++[1] | ++ | High[¥] | ++ | ++ |
| *Aspergillus* sp. | NM-17 | ++ | +++[1] | ++ | ++ | Moderate[¥] | +++ | ++ |
| *Aspergillus subalbidus* | NM-18 | +++ | +++[*][1] | +++[*][1] | ++ | High[¥] | +++ | ++ |
| *Aspergillus subalbidus* | NM-19 | +++ | +++[*][1] | +++[*][1] | ++ | High[¥] | +++ | ++ |
| *Aspergillus subalbidus* | NM-20 | +++ | +++[*][1] | +++[*][1] | ++ | High[¥] | +++ | ++ |
| *Aspergillus* sp. | NM-21 | +++ | +++[1] | ++ | ++ | Moderate | +++ | ++ |
| *Penicillium griseofulvum* | NM-22 | ++ | +++[1] | +++ | + | High[¥] | +++ | ++ |
| *Aspergillus subalbidus* | NM-23 | +++ | +++[*] | +++[*][1] | ++ | High[¥] | +++[*] | ++ |
| *Aspergillus subalbidus* | NM-24 | ++ | +++[*] | +++[*][1] | ++ | High[¥] | +++[*] | ++ |

**Key:**—(no growth), + (0–1.9 cm, slight growth), ++ (2.0–3.9 cm, moderate growth), +++ (>4.0 cm, abundant growth). Colony diameters were measured after 14 days of incubation. (*) indicates colony that shows growth of >4.0 cm after 7 days of incubation. ([1]) indicates NaCl concentration that gives optimal growth. ([¥]) indicates halophilic behavior of the isolates.

observed in the isolate NM-1; and for amylase production in isolates NM-2, NM-3 and NM-16, which had clear zones of 3.1–6.0 mm and were recorded as (++). High enzymatic activity for amylase production was shown by one isolate *Aspergillus* sp. NM-17, which showed a clear zone diameter above 6 mm and was scored as (+++). In addition, the EI was used to indicate the ability of each isolate to produce extracellular enzymes. Fungal isolates with an EI equal to or higher than 2 were classified as good candidates for enzyme production [34]. In terms of polyenzymatic activity, it was remarkable that some isolates could produce more than one enzyme (*Cladosporium tenuissimum* NM-2, *Penicillium citrinum* NM-3 and *P. griseofulvum* NM-22) (Table 4 and Fig 4).

## Screening for dsRNA mycoviruses

All 24 fungal isolates were screened for the presence of dsRNA mycoviruses. The dsRNA segments of three isolates (12.5%) were detected as bright and distinct bands in gel electrophoresis,. Two different dsRNA patterns were detected, ranging from 0.75 to 3.5 kb in genomic size (Fig 5). One isolate (*Fusarium equiseti* NM-8) presented two dsRNA fragments between 2.7 and3.5 kb, and two isolates (*Aspergillus subalbidus* NM-18 and NM-19) presented three dsRNA fragments between 0.75 and 1.9 kb. Enzymatic digestion was performed to confirm

Table 4. Summary of enzymatic activity of some of the studied fungal isolates. Only isolates positive for at least one substrate are shown.

| Strain identity | Isolate code | Amylase | | Cellulase | | Lipase | | Protease | |
|---|---|---|---|---|---|---|---|---|---|
| | | EA | EI | EA | EI | EA | EI | EA | EI |
| *Cladosporium tenuissimum* | NM-1 | - | - | - | - | - | - | ++ | 2.4 |
| *Cladosporium tenuissimum* | NM-2 | ++ | 2.0 | - | - | - | - | + | 1.6 |
| *Penicillium citrinum* | NM-3 | ++ | 2.2 | - | - | - | - | + | 1.5 |
| *Penicillium oxalicum* | NM-5 | + | 1.3 | - | - | - | - | - | - |
| *Penicillium oxalicum* | NM-6 | + | 1.2 | - | - | - | - | - | - |
| *Penicillium oxalicum* | NM-7 | + | 1.3 | - | - | - | - | - | - |
| *Aspergillus subalbidus* | NM-16 | ++ | 4.0 | - | - | - | - | - | - |
| *Aspergillus* sp. | NM-17 | +++ | 6.2 | - | - | - | - | - | - |
| *Aspergillus subalbidus* | NM-18 | - | - | - | - | - | - | + | 1.3 |
| *Aspergillus subalbidus* | NM-19 | + | 1.4 | - | - | - | - | - | - |
| *Aspergillus subalbidus* | NM-20 | - | - | - | - | - | - | + | 1.3 |
| *Penicillium griseofulvum* | NM-22 | + | 1.6 | - | - | - | - | + | 1.3 |
| *Aspergillus subalbidus* | NM-24 | - | - | - | - | - | - | + | 1.4 |

**Key:** Enzyme activity (EA);— = no activity, + = 0–3 mm, ++ = 3.1–6 mm, +++ = >6 mm.

their dsRNA identities. Primer sets were designed to yield PCR products from the coding regions of the RNA-dependent RNA polymerase (RdRP) and capsid protein (CP) of the partitivirus and chrysovirus previously described by Bhatti *et al.* [42]. However, no amplicon was generated.

## Antimicrobial assay

In primary screening, 17 out of 24 isolates (71%) showed antimicrobial activity against at least one pathogen using the culture broth filtrate (Fig 6 and Table 5). The presence of inhibition zones qualitatively indicated positive antimicrobial activity. Eleven out of 24 isolates (46%) had positive antimicrobial activity against Gram-positive and Gram-negative bacteria. Meanwhile, nine out of 24 isolates (38%; NM-3, NM-14, NM-15, NM-18, NM-19, NM-20, NM-22, NM-23 and NM-24) had positive antimicrobial activity against Gram-positive and Gram-negative bacteria; and yeast. Six isolates showed positive antimicrobial activity against bacteria,

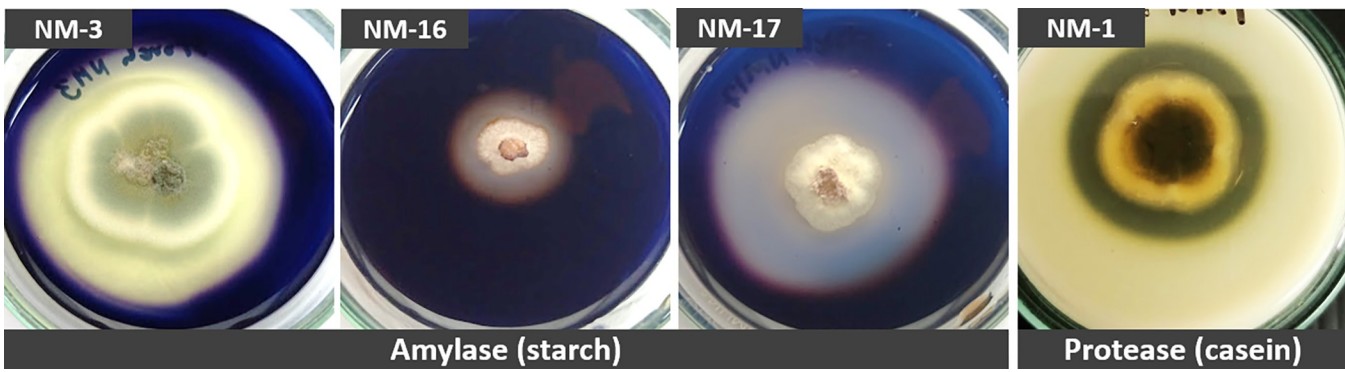

Fig 4. Enzymatic activity test. Four representative samples of the 13 fungal isolates which exhibited good extracellular enzymes activities.

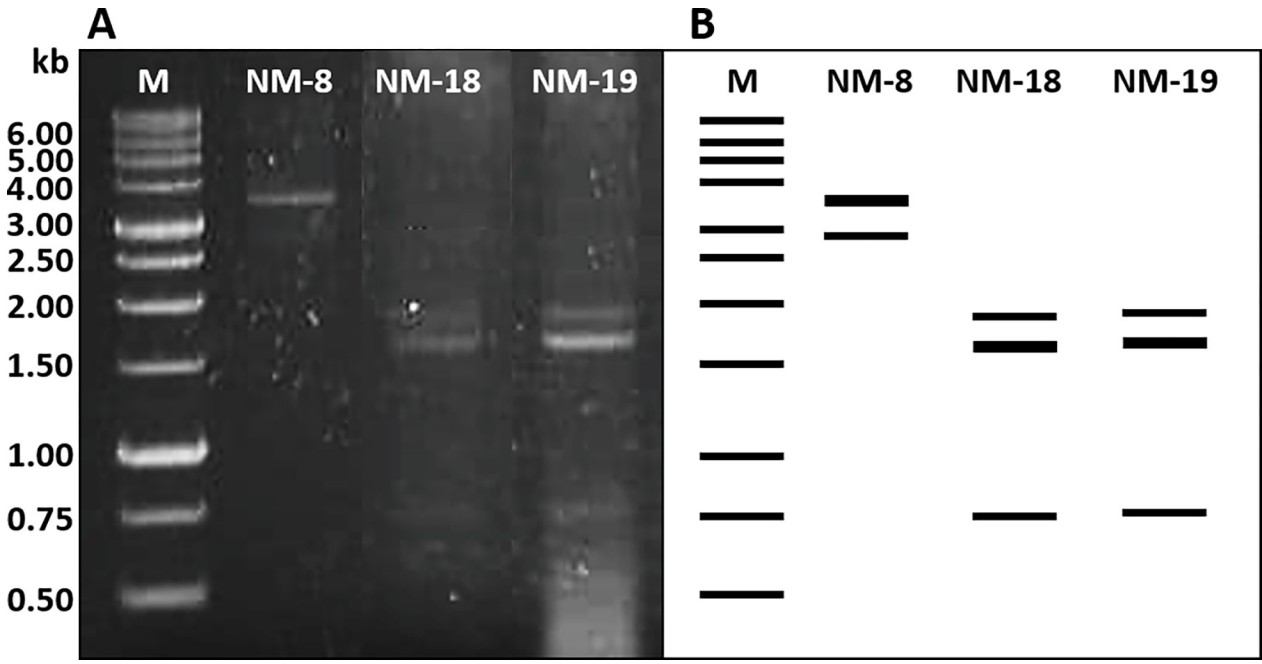

**Fig 5. Two putative dsRNA patterns of the 3 mycovirus infected strains.** A) Gel electrophoresis of isolated dsRNA patterns of fungal isolates. B) The dsRNA profiles displayed in lines.

yeast and filamentous fungi. *P. citrinum* NM-3 inhibited *Micrococcus luteus*, *Staphylococcus aureus*, *S. epidermidis*, MRSA, *Salmonella* Typhi, *Candida albicans*, *Aspergillus fumigatus* and *Microsporum canis*; and *A. subalbidus* NM-18, NM-19, NM-20 NM-23 and NM-24 inhibited *E. faecalis*, *M. luteus*, *S. aureus*, *S. epidermidis*, MRSA, *E. coli*, *S.* Typhi, *C. albicans* and *A. fumigatus*. It was remarkable that the *A. subalbidus* isolates showed the strongest antibacterial activity against *E. faecalis* and *S. aureus* and the strongest antifungal activity against *C. albicans*. Consideration of the antimicrobial activity of each fungal isolate against individual test organisms revealed that *A. subalbidus* NM-18, NM-19, NM-20 showed the most potent activity against *E. faecalis* and *S. aureus* while *Fusarium equiseti* NM-8 showed the most potent activity against *M. luteus* and MRSA. *A. subalbidus* NM-18, NM-19, NM-20, NM-23, NM-24 showed the strongest activity against *E. coli*, *S.* Typhi and *C. albicans* and *F. solani* NM-11 and *P. griseofulvum* NM-22 also showed strong activity against *C. albicans*. *A. subalbidus* NM-23 and NM-24 showed strongest activity against *A. fumigatus*. Results from this study indicated that fungi from solar salterns could be a good source of natural antimicrobial products.

Eight active fungal isolates (NM-3, NM-8, NM-15, NM-18, NM-19, NM-20, NM-23 and NM-24) were tested for antimicrobial activity at various concentrations (1–100 mg/mL) of ethanolic extracts of their culture broths and mycelia. The results demonstrated that ethanolic extracts of the culture broths showed better inhibition against several test pathogens than the mycelia extracts (S1 Table). In addition, the culture broth extracts produced larger inhibition zones than the mycelial extracts. Most of the extracts did not inhibit test organisms at 1 mg/mL. Exceptionally, 1 mg/mL ethanolic extracts of the culture broths from *P. citrinum* NM-3 and *F. equiseti* NM-8showed strong inhibitory activity against pathogenic bacteria that was comparable to the inhibitory activity of the positive controls. Overall, the culture broth and mycelial extracts exhibited strong antimicrobial activity against most test organisms at 10 and 100 mg/mL, respectively.

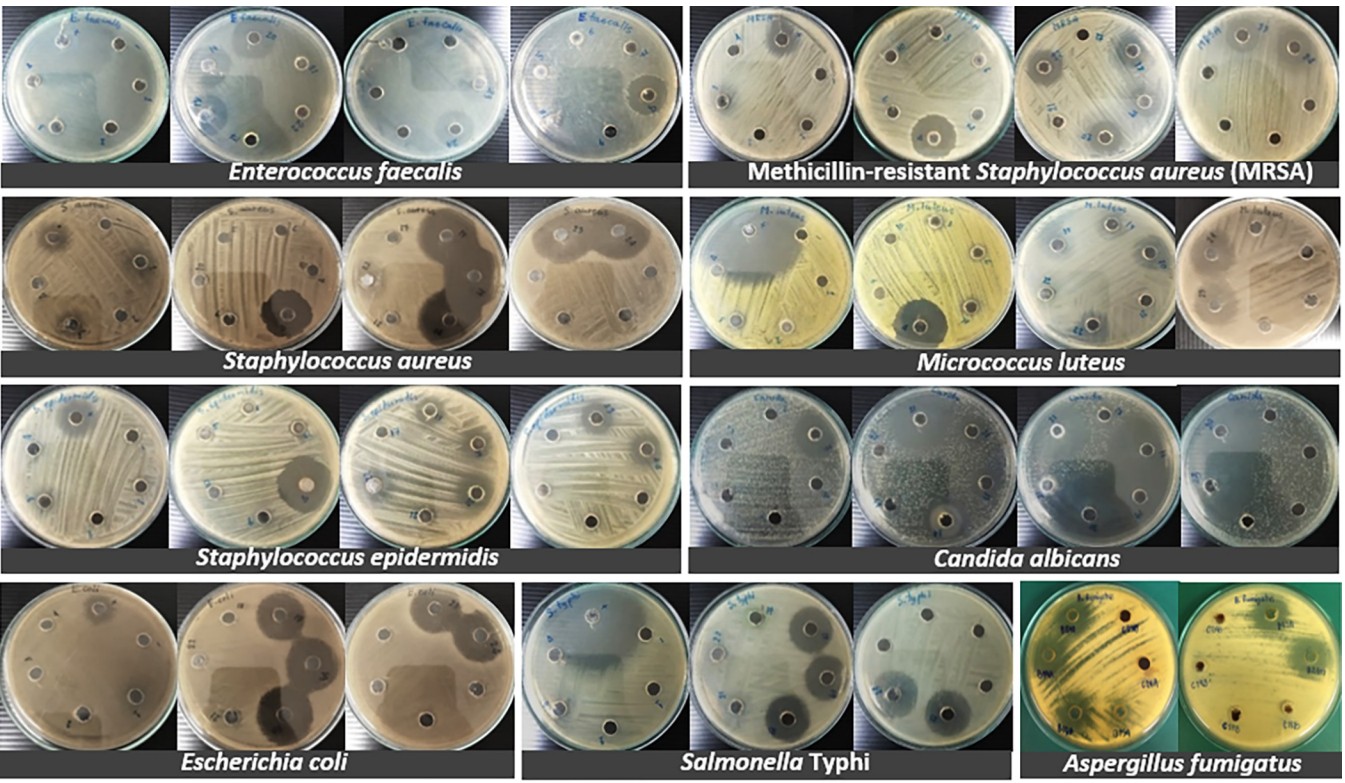

**Fig 6. Antimicrobial activity test.** Antimicrobial activity of active fungal isolates was determined by the agar well diffusion method. Culture broth filtrates of 80 µl were tested against pathogenic bacteria and fungi. Fungal filtrates exhibited large zones of inhibition for both Gram-positive (*E. faecalis*, *M. luteus* (ATCC9341), *S. aureus* (ATCC25923), *S. epidermidis*, methicillin-resistant *S. aureus* (MRSA)) and Gram-negative (*E. coli* (ATCC25922) and *S.* Typhi (ATCC19430)) bacteria, including pathogenic yeast (*C. albicans* (ATCC90028),) and fungi (*A. fumigatus* AF293).

## Discussion

Characterization of fungi isolated from man-made solar salterns in Pattani Province discovered extracellular enzymes, secondary metabolites and mycoviruses that may have potential for biotechnological applications. In addition, the diversity of halophilic and halotolerant fungi inhabiting this environment was revealed.

Morphological and molecular identification demonstrated seven different genera belonging to the phylum Ascomycota. These findings were consistent with the reports from Ndwigah *et al.* [5] and Salano *et al.* [43], who indicated the phylum Ascomycota as the dominant fungal group in salterns and saline environments. Fungal isolates were classified as *Aspergillus*, *Cladosporium*, *Curvularia*, *Diaporthe*, *Ectophoma*, *Fusarium* and *Penicillium*. Similar studies have reported the presence at diverse levels of the genera *Acremonium*, *Alternaria*, *Aspergillus*, *Cladosporium*, *Emericella*, *Eurotium*, *Fusarium*, *Penicillium* and *Wallemia* in hypersaline environments [2, 6–12, 44]. Our findings revealed that *A. subalbidus*, which is morphologically almost identical to *A. candidus*, was the most dominant species found in the soil of solar salterns, followed by *Penicillium* sp. Nayak *et al.* [2] and Gunde-Cimerman and Zalar [8] reported that *A. candidus* was abundant in hypersaline environments and distributed more locally. *F. equiseti*, previously found in hypersaline soils [45], was also isolated in our study. We also isolated the genera *Diaporthe* and *Ectophoma*. These are genera generally recognized as endophytic fungi, but their presence in terrestrial saline soils has not previously been reported. The presence of dominant strains from different fungal genera suggests an ability to develop strategies to adapt

**Table 5. Preliminary screening of antimicrobial activity of fungi isolated from salterns in Pattani Province, Thailand, was conducted using the agar well diffusion method.**

| Isolates | Zone of inhibition (mm) | | | | | | | | | | | Total no. |
|---|---|---|---|---|---|---|---|---|---|---|---|---|
| | Pathogenic bacteria | | | | | | | | Pathogenic fungi | | | |
| | Gram-positive | | | | | Gram-negative | | | Yeast | Mold | | |
| | EF* | ML | SA | SE | MRSA | EC | PA | ST | CA | AF | MC | |
| *Cladosporium tenuissimum* NM-1 | - | - | 12.5±0.4 [b] | - | - | 11.6±0.5 [a] | - | - | - | - | - | 2 |
| *Penicillium citrinum* NM-3** | - | 10.9±0.5 [a] | 15.2±0.8 [c] | 14.4±0.6 [c] | 14.8±1.3 [a] | - | - | 11.6±0.1 [a] | 25.7±0.7 [c] | 10.1+0.4 [a] | 9.3+0.6 [a] | 6 |
| *Penicillium oxalicum* NM-5 | - | 9.6±0.0 [a] | - | 9.9±0.0 [a] | - | - | - | - | - | - | - | 2 |
| *Penicillium oxalicum* NM-6 | - | 9.2±0.0 [a] | - | 8.5±0.0 [a] | - | - | - | - | - | - | - | 2 |
| *Penicillium oxalicum* NM-7 | - | 10.0±0.2 [a] | - | 9.9±0.2 [a] | - | - | - | - | - | - | - | 2 |
| *Fusarium equiseti* NM-8** | 23.3±0.3[a] | 29.7±0.6 [g] | 26.9±1.0 [f] | 28.9±1.8 [e] | 26.1±0.9 [c] | - | - | - | 25.0±2.1 [c] | 15.8+1.5 [c] | 10.0+0.5 [a] | 6 |
| *Fusarium solani* NM-11 | - | - | 17.6±0.7 [d] | 16.7±0.5 [d] | - | - | - | - | 32.1±1.0 [d] | - | - | 3 |
| *Aspergillus niger* NM-14** | - | 11.4±0.5 [b] | 13.2±0.2 [b] | 9.4±0.0 [a] | - | - | 8.5±0.5 [a] | - | 19.6±0.9 [b] | - | - | 5 |
| *Aspergillus nomius* NM-15** | - | - | 10.6±0.0 [a] | - | - | - | 10.0±0.0 [a] | 10.6±0.0 [a] | 17.1±0.8 [a] | - | - | 4 |
| *Aspergillus subalbidus* NM-16** | - | 12.0±0.4 [b] | 13.5±1.0 [b] | 11.5±0.5 [b] | - | - | - | - | 16.1±0.5 [a] | - | - | 4 |
| *Aspergillus subalbidus* NM-18** | 24.7±0.7 [a] | 19.7±0.3 [d] | 35.7±1.5 [h] | 15.2±0.4 [c] | 14.5±0.6 [a] | 27.8±1.7 [b] | - | 24.5±0.4 [b] | 33.3±1.0 [d] | 11.9+0.9 [b] | - | 8 |
| *Aspergillus subalbidus* NM-19** | 24.8±0.9 [a] | 17.5±0.5 [c] | 35.1±0.4 [h] | 14.6±0.0 [c] | 15.9±1.1 [a] | 26.9±0.8 [b] | - | 23.7±0.6 [b] | 35.8±1.9 [e] | 12.3+1.1 [b] | - | 8 |
| *Aspergillus subalbidus* NM-20** | 24.2±0.3 [a] | 20.4±0.4 [e] | 33.9±0.4 [g] | 28.0±0.5 [e] | 17.5±1.1 [b] | 28.5±1.5 [b] | - | 24.0±0.9 [b] | 33.4±0.5 [d] | 11.9+0.9 [b] | - | 8 |
| *Aspergillus* sp. NM-21 | - | - | - | - | - | - | 11.0±0.0 [a] | - | - | - | - | 1 |
| *Penicillium griseofulvum* NM-22** | - | 15.7±09 [c] | 19.7±0.0 [e] | 17.5±0.6 [d] | 18.3±0.4 [b] | - | - | 10.6±0.5 [a] | 32.7±2.1 [d] | - | - | 6 |
| *Aspergillus subalbidus* NM-23** | 21.1±0.0 [b] | 25.3±0.5 [f] | 33.6±0.5 [g] | 17.0±0.0 [d] | 18.0±0.0 [b] | 28.9±0.5 [b] | - | 25.1±0.2 [b] | 32.6±0.6 [f] | 19.9+0.8 [c] | - | 8 |
| *Aspergillus subalbidus* NM-24** | 21.5±1.1 [b] | 25.5±0.2 [f] | 33.3±0.4 [g] | 15.5±0.2 [c] | 17.8±0.5 [b] | 27.5±0.2 [b] | - | 25.5±1.5 [b] | 36.9±1.2 [g] | 21.6+0.1 [d] | - | 8 |
| Total no. | 6 | 13 | 13 | 14 | 8 | 6 | 3 | 8 | 12 | 3 | 3 | |
| Positive control | EF* | ML | SA | SE | MRSA | EC | PA | ST | CA | AF | MC | |
| Chloramphenicol (50 μg/mL) | 55.1±1.2 | 59.1±1.5 | 17.0±0.5 | 20.0±0.7 | 15.1±0.2 | 17.1±0.3 | 39.0±0.5 | 55.1±2.0 | | | | |
| Amphotericin B (50 μg/mL) | | | | | | | | | 22.0±1.5 | 20.0±0.7 | 24.0±0.2 | |
| Vancomycin (20 μg/mL) | 13.7±1.9 | 18.5±0.1 | 14.7±0.5 | 13.1±0.6 | 12.9±0.3 | | | | | | | |

(*Continued*)

**Table 5.** (Continued)

| Isolates | Zone of inhibition (mm) | | | | | | | | | | | Total no. |
|---|---|---|---|---|---|---|---|---|---|---|---|---|
| | Pathogenic bacteria | | | | | | | | Pathogenic fungi | | | |
| | Gram-positive | | | | | Gram-negative | | | Yeast | Mold | | |
| | EF* | ML | SA | SE | MRSA | EC | PA | ST | CA | AF | MC | |
| Gentamicin (20 μg/mL) | | | | | | 19.7±1.7 | 14.2±0.5 | 20.7±1.2 | | | | |
| Amphotericin B (20 μg/mL) | | | | | | | | | 19.3±0.6 | | | |
| Miconazole (20 μg/mL) | | | | | | | | | | 13.9±1.9 | 12.6±0.3 | |

*EF: *E. faecalis*; ML: *M. luteus* (ATCC9341); SA: *S. aureus* (ATCC25923); SE: *S. epidermidis*; MRSA: methicillin-resistant *S. aureus* (MRSA), EC: *E. coli* (ATCC25922), PA: *P. aeruginosa* (ATCC27853); ST: *S.* Typhi (ATCC19430); CA: *C. albicans* (ATCC90028), AF: *A. fumigatus* (AF293); and MC: *M. canis*. The same superscript letter[(a-h)] indicates no significant difference ($p > 0.05$) between zone of inhibition values from different endophytic fungi against each test organism (same column).

**Number of most active fungal isolates that inhibit more than 4 test microorganisms with inhibition zones of $\geq$ 10 mm. -: no activity.

to the extreme hypersaline conditions of the solar salterns. These strategies include the production of extremolytes and extremozymes [16] or cellular accumulations of K$^+$ ions and osmolytes [17, 18]. Other strategies are, for instance, changes in ion transport or plasma membrane fluidity which play a crucial role in adaptation to high salt concentrations [8]. Further studies of the biotechnological applications of these fungi would be of interest owing to their ability to adapt to saline environments.

The biotechnological interest of the fungal isolates was investigated by testing their ability to grow at different NaCl concentrations and to produce extracellular enzymes and secondary metabolites. All the studies fungal isolates could grow across all tested salt concentrations (0–20% NaCl). The halotolerance test revealed that the majority of the isolates are halotolerant. Halophilic behavior (optimum growth at 10% NaCl) was observed from fungi belonging to the genera *Aspergillus*, *Fusarium* and *Penicillium*, while the other genera were halotolerant. In this study, *Cladosporium* and *A. subalbidus* were the most halotolerant fungi because they withstood the highest NaCl concentration (20% NaCl). It emerged that most of the halotolerant isolates were melanized fungi. These included *Cladosporium*, *Curvularia* and *Ectophoma*. The dark-pigmented character of these fungi was reported to be an essential factor for stress tolerance [17, 46, 47]. The optimal temperature required for all fungal growth was 28˚C but some isolates also showed growth at 40˚C. It has been shown that most halophilic microorganisms grow optimally at temperatures between 25 and 45˚C [48].

The enzymatic activity of fungal isolates revealed their ability to produce extracellular enzymes. Thirteen out of 24 isolates showed the ability to produce proteases and amylases, meanwhile the production of cellulases and lipases was not detected. Certain isolates could be of interest for industrial applications. *C. tenuissimum* NM-1 could be used for protease production. *P. citrinum* NM-3, *A. subalbidus* NM-16 and *Aspergillus* sp. NM-17 could perhaps be applied in amylase production because they exhibited an EI higher than 2 [34]. The strain that presented the highest EI (6.2) was *Aspergillus* sp. NM-17. Our results supported the findings of Dendouga and Belhamra [12] and Chamekh *et al.* [45], who reported high production of protease from *Cladosporium* and high production of amylase from *Aspergillus* and *Penicillium*. In addition, their ability to tolerate salt stress is a characteristic that makes them good candidates for halophilic enzyme production.

Three dsRNA mycoviruses were detected among the isolates: *F. equiseti* NM-8 and *A. subalbidus* NM-18 and NM-19. They contained 2–3 genomic segments ranging in size from 0.75 to 3.5 kb, comparable in size to multipartite dsRNA fragments from *Aspergillus* mycoviruses of

the Partitiviridae and Chrysoviridae families [49, 50]. PCR amplification was performed to determine whether the mycoviruses observed in this study were partitiviruses or chrysoviruses, but no PCR product was generated. Although the dsRNA molecules obtained were comparable in size and genomic segmentation to the partitivirus and chrysovirus in the study of Bhatti *et al.* [42], they were not the same This also indicated that detected dsRNA mycoviruses could represent members of yet uncharacterized Partitiviridea and Chrysoviridae or belong to another virus family. Notably, the fungi harboring dsRNA mycoviruses in this study were halophilic and were positive for enzyme and antimicrobial activities. These properties could contribute to the presence of mycoviruses in the fungal hosts. Several studies have reported the advantages mycoviruses confer on fungi, for example, increased thermal tolerance in the endophytic fungus *C. protuberata* [22], and induced metabolite production in *M. oryzae* [23]. In addition, mycoviruses could theoretically be used as biological control agents against phytopathogenic fungi [51]. To identify potential biotechnological applications of mycoviruses of halophilic fungi, dsRNA molecules of the mycoviruses must be further sequenced and characterized.

Fungi from saline environments possess the potential to produce several secondary metabolites, including antimicrobial agents. In this study, 71% of the isolates showed antimicrobial activity against at least one pathogen. Antimicrobial production was noted for six isolates, which showed positive antimicrobial activity against bacteria, yeast and filamentous fungi. The isolates *P. citrinum* NM-3 and *A. subalbidus* NM-18, NM-19, NM-20, NM-23 and NM-24) proved to be active against the Gram-positive bacteria *E. faecalis*, *M. luteus*, *S. aureus*, *S. epidermidis* and MRSA, the Gram-negative bacteria *E. coli* and *S.* Typhi, the yeast *C. albicans* and also the human pathogenic fungus *A. fumigatus*. It is known that various polyketides, including antibiotics (penicillins), extrolites and penicillic acids, are produced by halophilic strains of *Penicillium* [52] and that, antimicrobial compounds such as β-lactam antibiotics from *P. chrysogenum* and other *Penicillium* species inhibit most Gram-positive bacteria [53]. In the light of this existing knowledge, the further characterization and purification of the active metabolites produced by the above six strains of *P. citrinum* and *A. subalbidus* are being investigated. The expectation is that the discovery of fungi in saline environments could further the exploration of new fungal strains and lead to innovative biotechnological and industrial applications involving their enzymes, metabolites and mycoviruses.

## Conclusion

Our studies presented culturable halophilic and halotolerant fungi isolated from man-made solar salterns in Pattani Province, Thailand. A total of 24 fungal isolates were discovered and phylogenetically classified into seven genera belonging to *Aspergillus*, *Cladosporium*, *Curvularia*, *Diaporthe*, *Ectophoma*, *Fusarium* and *Penicillium*. Growth under varying salt concentrations confirmed the halophilic and halotolerant nature of some isolates. This study demonstrated the ability of some halophilic fungi to produce proteases and amylases. In addition, some fungal isolates were shown to harbor mycoviruses. Most of the isolates produced antimicrobial metabolites which displayed inhibitory effects on Gram-positive and Gram-negative bacteria, including human pathogenic fungi. The ability of halophilic fungi to produce extracellular enzymes and antimicrobial agents, including properties of their mycoviruses should be a subject of further investigation. Notably, the exploration of fungal strains inhabiting saline environments could open up new biotechnological and industrial applications.

## Supporting information

**S1 Fig. Halotolerance test of the isolated fungi.** Results were evaluated based on colony diameter after 14 days of growth at 28˚C under various NaCl concentrations

(0–20%(w/v)).
(TIF)

**S1 Table. Antimicrobial activity of fungi isolated from saltern farms in Pattani Province.**
Antimicrobial activity of ethanolic extracts of the culture broths and mycelia of eight active isolates using the agar well diffusion method.
(PDF)

**S1 Raw image.**
(PDF)

## Author Contributions

**Conceptualization:** Lakkhana Kanhayuwa Wingfield.

**Data curation:** Lakkhana Kanhayuwa Wingfield.

**Methodology:** Lakkhana Kanhayuwa Wingfield, Ninadia Jitprasitporn, Nureeda Che-alee.

**Supervision:** Lakkhana Kanhayuwa Wingfield.

**Writing – original draft:** Lakkhana Kanhayuwa Wingfield.

**Writing – review & editing:** Lakkhana Kanhayuwa Wingfield.

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
