## [Decision Letter · Decision Letter 0]

17 Jun 2022

PONE-D-22-14572Isolation and characterization of halophilic and halotolerant fungi from man-made solar salterns in Pattani province, ThailandPLOS ONE

Dear Dr. Wingfield,

Thank you for submitting your manuscript to PLOS ONE. After careful consideration, we feel that it has merit but does not fully meet PLOS ONE’s publication criteria as it currently stands. Therefore, we invite you to submit a revised version of the manuscript that addresses the points raised during the review process.

ACADEMIC EDITOR: The concerns addressed by the reviewers were more related the English language presentation and the figure legend details and images in some cases and the selected references to cite rather than scientific issues so please focus on responding to these concerns in the revised manuscript. 

We look forward to receiving your revised manuscript.

Kind regards,

William C. Nierman, Ph.D.

Academic Editor

PLOS ONE

Journal Requirements:

Reviewers' comments:

Reviewer's Responses to Questions

**Comments to the Author**

1. Is the manuscript technically sound, and do the data support the conclusions?

Reviewer #1: Yes

Reviewer #2: Yes

2. Has the statistical analysis been performed appropriately and rigorously? 

Reviewer #1: Yes

Reviewer #2: Yes

3. Have the authors made all data underlying the findings in their manuscript fully available?

Reviewer #1: Yes

Reviewer #2: Yes

4. Is the manuscript presented in an intelligible fashion and written in standard English?

Reviewer #1: No

Reviewer #2: No

5. Review Comments to the Author

Reviewer #1: The manuscript covers the novel aspects in the niche of extremophilic fungi, which is appreciated and is recommended for publication after following few points are satisfied in revision.

1. The overall English of the manuscript needs extensive revision and improvement.

2. The literature review in introduction as well as paper comparisons with previous publications are citing decade old citations which better needs to be from last 5 years or at most from 7-8 years older. Specifically the halophilic fungal diversity and applications needs to be updated.

3. Try to discuss that how these fungi falls in halophilic category by consulting publications such as Hypersaline habitats and halophilic microorganisms, 2016, Maejo International Journal of Science and Technology 10(3):330-345.

4. For the unit of ml or ul, use capital L.

5. Better add a separate conclusions having future perspective of your research.

Reviewer #2: General

The manuscript requires a major rewriting to upgrade the quality of the English language presentation.

The citations in many instances for very old. More recent citations would strengthen the presentation of the manuscript.

For the first presentation of the genus and species of an organism, the genus and species should be spelled out. For subsequent occasions the first letter of the genus name can be abbreviated, i.e. Escherichia coli(first presentation) and E. coli (subsequent presentations).

When using "et al." in the manuscript text or reference list, it should be in italics.

Specific comments on the manuscript.

Pg. 4 line 94. The Fig. 1 legend requires more details. The windows should be labeled with an upper case letter designation (i.e. A, B,... ) and text should be provided in the legend to describe what is in each image.

Pg. 4 lines 101 and 102. The two analytical procedures require literature references as was done for the Tecator DS-6 Digester line 103.

Pg. 7 line 163. Is the enzymatic index a standard approach to reporting a quantitative measure of enzyme activity. If so it requires a reference.

Pg. 10 line 215. The Fig. 2 legend does not provide enough detail - see the line 94 comment on Fig. 1.

Pg. 15 line 318. The labes "A" and "B" are not indicated on Fig. 5.

Pg. 16 line 354. Insufficient details is provided in the Fig. 6 legend.

Pg. 19 Line numbers are discontinued with the start of the Discussion section.

6. PLOS authors have the option to publish the peer review history of their article (what does this mean?). If published, this will include your full peer review and any attached files.

Reviewer #1: **Yes: **Imran Ali

Reviewer #2: No

---

## [Author Response · Author response to Decision Letter 0]

3 Jan 2023

5. Review Comments to the Author

Reviewer #1: 

1. The overall English of the manuscript needs extensive revision and improvement.

## The proposed revision has been made.

2. The literature review in introduction as well as paper comparisons with previous publications are citing decade old citations which better needs to be from last 5 years or at most from 7-8 years older. Specifically, the halophilic fungal diversity and applications needs to be updated.

## I understand that some citations presented in the literature review and discussion of this manuscript are not very up-to-date. After reviewing and researching for more recent publications (2018-2022) on halophilic fungi, most of the recent works have been focused on cell biology, physiological response (particularly stress response, and mechanisms of cellular and molecular adaptation to salinity), and in-depth investigation on metabolic potential of halophilic fungi. We assume they are rather irrelevant to our research scope which focuses on fungal diversity and their biotechnological potential (initial investigation). 

We also try to cover the most current research in the field of halophilic fungal diversity by including both early and very new research. However, few recent publications have described fungal diversity in hypersaline environments, for example; Chamekh et al., 2019. (Isolation, identification and enzymatic activity of halotolerant and halophilic fungi from the Great Sebkha of Oran in northwestern of Algeria. Mycobiology. 47(2):230-241), Rashmi et al., 2018. (Fungal diversity of the hypersaline Inland Sea in Qatar. Botanica Marina. 61(6), 595-609., Chung et al., 2019. (Fungi in salterns. J Microbiol. 57(9), 717-724) and Dendouga et al., 2022. (Screening of halotolerant microfungi isolated from hypersaline soils of Algerian Sahara for production of hydrolytic enzymes. J of Biol Res. 95:10167). In addition, up-to-date publication on antimicrobial activity of Penicillium was published (Orfali et al. Anti-microbial activity of dihydroisocoumarin isolated from Wadi Lajab sediment-derived fungus Penicillium chrysogenum: In vitro and In silico study. Molecules. 2022; 27: 3630. https://doi.org/10.3390/molecules27113630). These publications are cited in our manuscripts (## Line 367-670, 403-408, 434-436).

As suggested, more recent citations would strengthen the presentation of the manuscript. I strongly believe this logic. However, as far as I am concerned, some recent publications on halophilic fungi still cite old publications in order to support their findings/hypotheses, for example, Pupo et al., 2021. (Osmolyte signatures for the protection of Aspergillus sydowii cells under halophilic conditions and osmotic shock. J Fungi. 7, 414. https://doi.org/10.3390/jof7060414) as well as publications aforementioned. 

In our case, some old references are used because they need to be able to cover the materials. We cited old publications on a case-by-case basis. For example, the methods we followed belong to the original publications which remain applicable nowadays (identification keys of fungi, primer set for PCR amplification, enzyme and antimicrobial assays). Result interpretation, for example, enzyme activity was evaluated by an enzymatic index described by original work of Hankin (900 citations). 

In term of diversity of fungal halophiles (either stated on introduction or discussion), we sometimes compare our results or cite literature to decade-old publications because they are relevant to the point we are trying to make. For example, we demonstrated fungal genera found from our sampling sites. These included A. subalbidus and Fusarium which were also reported from two old publications (2006 and 2014). In literature review, we include old citations to introduce fungal genera that have been found from hypersaline environments so far. Similarly, description of characteristics of hypersaline soils, groups of halophiles, stress response to salinity in fungi we cited here were from the original publications. Overall, in my opinion, much more relevant than publication year is the content of the publications cited that cover relevant literature/discussion. We definitely try to cover the most current research in the field of halophilic fungal diversity by including both early and very new research as stated. However, if you have any additional suggestion regarding this matter, I would be very appreciated to correct points you suggest. 

3. Try to discuss that how these fungi fall in halophilic category by consulting publications such as Hypersaline habitats and halophilic microorganisms, 2016, Maejo International Journal of Science and Technology 10(3):330-345.

## The proposed correction has been made in the revised version of our manuscript.

4. For the unit of ml or ul, use capital L.

## The proposed corrections have been made in the revised version of our manuscript.

5. Better add a separate conclusion having future perspective of your research.

## The proposed correction has been made in the revised version of our manuscript (line 442-454).

Reviewer #2: General

The manuscript requires a major rewriting to upgrade the quality of the English language presentation. 

## The proposed correction has been made in the revised version of our manuscript.

The citations in many instances for very old. More recent citations would strengthen the presentation of the manuscript.

## I understand that some citations presented in the literature review and discussion of this manuscript are not very up-to-date. After reviewing and researching for more recent publications (2018-2022) on halophilic fungi, most of the research works have been focused on cell biology, physiological response (particularly stress response, and mechanisms of cellular and molecular adaptation to salinity), and in-depth investigation on metabolic potential of halophilic fungi. We assume they are rather irrelevant to our scope of research which focuses on fungal diversity and their biotechnological potential (initial investigation). 

We also try to cover the most current research in the field of halophilic fungal diversity by including both early and very new research. However, few recent publications have described fungal diversity in hypersaline environments, for example; Chamekh et al., 2019. (Isolation, identification and enzymatic activity of halotolerant and halophilic fungi from the Great Sebkha of Oran in northwestern of Algeria. Mycobiology. 47(2):230-241), Rashmi et al., 2018. (Fungal diversity of the hypersaline Inland Sea in Qatar. Botanica Marina. 61(6), 595-609., Chung et al., 2019. (Fungi in salterns. J Microbiol. 57(9), 717-724) and Dendouga et al., 2022. (Screening of halotolerant microfungi isolated from hypersaline soils of Algerian Sahara for production of hydrolytic enzymes. J of Biol Res. 95:10167). In addition, up-to-date publication on antimicrobial activity of Penicillium was published (Orfali et al. Anti-microbial activity of dihydroisocoumarin isolated from Wadi Lajab sediment-derived fungus Penicillium chrysogenum: In vitro and In silico study. Molecules. 2022; 27: 3630. https://doi.org/10.3390/molecules27113630). These publications are cited in our manuscripts (## Line 367-670, 403-408, 434-436).

As suggested, more recent citations would strengthen the presentation of the manuscript. I strongly believe this logic. However, as far as I am concerned, some recent publications on halophilic fungi still cite old publications in order to support their findings/hypotheses, for example, Pupo et al., 2021. (Osmolyte signatures for the protection of Aspergillus sydowii cells under halophilic conditions and osmotic shock. J Fungi. 7, 414. https://doi.org/10.3390/jof7060414) as well as publications aforementioned. 

In our case, some old references are used because they need to be able to cover the materials. We cited old publications on a case-by-case basis. For example, the methods we followed belong to the original publications which remain applicable nowadays (identification keys of fungi, primer set for PCR amplification, enzyme and antimicrobial assays). Result interpretation, for example, enzyme activity was evaluated by an enzymatic index described by original work of Hankin (900 citations). 

In term of diversity of fungal halophiles (either stated on introduction or discussion), we sometimes compare our results or cite literature to decade-old publications because they are relevant to the point we are trying to make. For example, we demonstrated fungal genera found from our sampling sites. These included A. subalbidus and Fusarium which were also reported from two old publications (2006 and 2014). In literature review, we include old citations to introduce fungal genera that have been found from hypersaline environments so far. Similarly, description of characteristics of hypersaline soils, groups of halophiles, stress response to salinity in fungi we cited here were from the original publications which are undoubtedly old. Overall, in my opinion, much more relevant than publication year is the content of the publications cited that cover relevant literature/discussion. We definitely try to cover the most current research in the field of halophilic fungal diversity by including both early and very new research as stated. However, if you have any additional suggestion regarding this matter, I would be very appreciated to correct points you suggest. 

For the first presentation of the genus and species of an organism, the genus and species should be spelled out. For subsequent occasions the first letter of the genus name can be abbreviated, i.e., Escherichia coli (first presentation) and E. coli (subsequent presentations).

## The proposed corrections have been made in the revised version of our manuscript (line 26, 66, 173-176, 259-261, 266, 326-327 and discussion).

When using "et al." in the manuscript text or reference list, it should be in italics.

## The proposed corrections have been made in the revised version of our manuscript (line 113, 313 and discussion).

Specific comments on the manuscript.

Pg. 4 line 94. The Fig. 1 legend requires more details. The windows should be labeled with an upper case letter designation (i.e. A, B,... ) and text should be provided in the legend to describe what is in each image.

## The proposed corrections have been made in the revised version of our manuscript (lone 94).

Pg. 4 lines 101 and 102. The two analytical procedures require literature references as was done for the Tecator DS-6 Digester line 103.

## The proposed corrections have been made in the revised version of our manuscript (line 101).

Pg. 7 line 163. Is the enzymatic index a standard approach to reporting a quantitative measure of enzyme activity. If so it requires a reference.

## Yes, the enzymatic index stated in this manuscript is a standard approach to reporting a quantitative measure of enzyme activity as described by Hankin (1975).

The proposed correction has been made in the revised version of our manuscript (line 164).

Pg. 10 line 215. The Fig. 2 legend does not provide enough detail - see the line 94 comment on Fig. 1.

## The proposed correction has been made in the revised version of our manuscript (line 212).

Pg. 15 line 318. The labes "A" and "B" are not indicated on Fig. 5.

## The proposed correction has been made in the revised version of our manuscript (on Fig 5).

Pg. 16 line 354. Insufficient details is provided in the Fig. 6 legend.

## The proposed correction has been made in the revised version of our manuscript (line 349).

Pg. 19 Line numbers are discontinued with the start of the Discussion section.

## The proposed correction has been made in the revised version of our manuscript.

6. We note that the grant information you provided in the ‘Funding Information’ and ‘Financial Disclosure’ sections do not match. When you resubmit, please ensure that you provide the correct grant numbers for the awards you received for your study in the ‘Funding Information’ section.

## The proposed correction has been made.

7. PLOS ONE now requires that authors provide the original uncropped and unadjusted images underlying all blot or gel results reported in a submission’s figures or Supporting Information files. This policy and the journal’s other requirements for blot/gel reporting and figure preparation are described in detail at https://journals.plos.org/plosone/s/figures#loc-blot-and-gel-reporting-requirements and https://journals.plos.org/plosone/s/figures#loc-preparing-figures-from-image-files. When you submit your revised manuscript, please ensure that your figures adhere fully to these guidelines and provide the original underlying images for all blot or gel data reported in your submission. 

## The proposed correction has been made.

---

## [Editor Report · Decision Letter 1]

23 Jan 2023

PONE-D-22-14572R1Isolation and characterization of halophilic and halotolerant fungi from man-made solar salterns in Pattani province, ThailandPLOS ONE

Dear Dr. %Wingfield%,

Thank you for submitting your manuscript to PLOS ONE. After careful consideration, we feel that it has merit but does not fully meet PLOS ONE’s publication criteria as it currently stands. Therefore, we invite you to submit a revised version of the manuscript that addresses the points raised during the review process. Most of the concerns raised by the reviewers in their comments have been well address by the authors in the submitted revised manuscript. Two comments remain inadequately addressed,  that the manuscript would benefit by further enhancing the English presentation as requested by both reviewers, and instances of improper usage of abbreviated genus names remain uncorrected in the manuscript. Suggestions for addressing these issues are provided.

We look forward to receiving your revised manuscript.

Kind regards,

William C. Nierman, Ph.D.

Academic Editor

PLOS ONE

Journal Requirements:

Additional Editor Comments (if provided):

Most of the concerns raised by the reviewers in their comments have been well address by the authors in their submitted revised manuscript. Two comments remain inadequately addressed, the manuscript would benefit by further enhancing the English presentation requested by both reviewers, and instances of improper usage of abbreviated genus names remain uncorrected in the manuscript. To assist in preparing a further revised version of the manuscript the following suggestions are offered for consideration by the authors. Line numbers used are those from the Track Changes version of the revised manuscript.

Ln 15-16 replace "physiologistic characteristics" with "levels of salinity"

Ln 18 replace "test" with "testing"

Ln 24 replace "S." with "Staphylococcus"

Ln 26 replace "A." "Aspergillus"

28 replace "interests" with "applications"

Ln 33 after "unicellular" add "yeast"

after "multicellular" add "hyphi"

replace "consisting" with "that grow as"

Ln 36 delete "in the"

Ln 37 after"3.5%' add ", the salinity"

Ln 40 "psu" needs to be defined

Ln 66 replace "C." with "Curvularia"

Ln 80 after “,” add “and with potential”

Ln 82 replace “ interests” with “applications”

Ln 287 after “isolates” add “by the assays employed in the study.”

Ln 385 replace “interest” with “adaptability and production potential og useful enzymes and other biologicals.”

Ln 396 replace “evident” with “reported”

Ln 399 replace “meanwhile” with “while”

Ln 402 replace “index” with “indices”

Ln 414 replace “It could be explained that” with “It is likely that”

Ln 432 After “and also” add “the”

Ln 438 replace “exploration” with “discovery”

Ln 439 after “ fungal strains” delete all and replace with “ and expand the quest for useful enzymes and metabolites to include halotolerant varieties. Mycoviruses may also find biotech applications in modulating the properties of halophilic and halotolerant fungi”

Ln 448 replace “showed” with “were shown”

Ln 450 after “bacteria” add “and” and delete “including”

Ln 451-452 after “mycoviruses” add “are likely to be productive subjects for further investigation.”

Ln 454 replace “interests” with ”applications”
---

## [Author Response · Author response to Decision Letter 1]

25 Jan 2023

Journal Requirements:

# We have reviewed the reference list and no retracted paper was cited in our manuscript.

Additional Editor Comments (if provided):

Most of the concerns raised by the reviewers in their comments have been well address by the authors in their submitted revised manuscript. Two comments remain inadequately addressed, the manuscript would benefit by further enhancing the English presentation requested by both reviewers, and instances of improper usage of abbreviated genus names remain uncorrected in the manuscript. To assist in preparing a further revised version of the manuscript the following suggestions are offered for consideration by the authors. Line numbers used are those from the Track Changes version of the revised manuscript.

# English presentation of our manuscript has been revised and improved by language editorials from the Publication Clinic Service at Prince of Songkla University. All changes are shown in blue typeface/highlight in our Revised Manuscript with Track Changes. Improper usage of abbreviated genus names was corrected as suggested. All suggestions were applied in our manuscript.

---

## [Editor Report · Decision Letter 2]

30 Jan 2023

Isolation and characterization of halophilic and halotolerant fungi from man-made solar salterns in Pattani Province, Thailand

PONE-D-22-14572R2

Dear Dr. Wingfield,

We’re pleased to inform you that your manuscript has been judged scientifically suitable for publication and will be formally accepted for publication once it meets all outstanding technical requirements.

Kind regards,

William C. Nierman, Ph.D.

Academic Editor

PLOS ONE
---

## [Editor Report · Acceptance letter]

3 Feb 2023

PONE-D-22-14572R2 

Isolation and characterization of halophilic and halotolerant fungi from man-made solar salterns in Pattani Province, Thailand 

Dear Dr. Wingfield:

I'm pleased to inform you that your manuscript has been deemed suitable for publication in PLOS ONE. Congratulations! Your manuscript is now with our production department. 

Kind regards, 

on behalf of

Dr. William C. Nierman 

Academic Editor

PLOS ONE